# Usability and Acceptability of a Novel Secondary Prevention Initiative Targeting Physical Activity for Individuals after a Transient Ischaemic Attack or “Minor” Stroke: A Qualitative Study

**DOI:** 10.3390/ijerph17238788

**Published:** 2020-11-26

**Authors:** Neil Heron, Sean R. O’Connor, Frank Kee, Jonathan Mant, Margaret E. Cupples, Michael Donnelly

**Affiliations:** 1Centre for Public Health, Queen’s University, CPH QUB, Belfast BT12 6BA, UK; f.kee@qub.ac.uk (F.K.); m.cupples@qub.ac.uk (M.E.C.); michael.donnelly@qub.ac.uk (M.D.); 2School of Primary, Community and Social Care, Keele University, Newcastle ST5 5BG, UK; s.oconnor@qub.ac.uk; 3Primary Care Unit, Department of Public Health and Primary Care, University of Cambridge, Cambridge CB2 0SR, UK; jm677@medschl.cam.ac.uk

**Keywords:** transient ischaemic attack, “minor” stroke, physical activity, secondary prevention, cardiovascular risk

## Abstract

Behavioural interventions that address cardiovascular risk factors such as physical inactivity and hypertension help reduce recurrence risk following a transient ischaemic attack (TIA) or “minor” stroke, but an optimal approach for providing secondary prevention is unclear. After developing an initial draft of an innovative manual for patients, aiming to promote secondary prevention following TIA or minor stroke, we aimed to explore views about its usability and acceptability amongst relevant stakeholders. We held three focus group discussions with 18 participants (people who had experienced a TIA or minor stroke (4), carers (1), health professionals (9), and researchers (4). Reflexive thematic analysis identified the following three inter-related themes: (1) relevant information and content, (2) accessibility of format and helpful structure, and (3) strategies to optimise use and implementation in practice. Information about stroke, medication, diet, physical activity, and fatigue symptoms was valued. Easily accessed advice and practical tips were considered to provide support and reassurance and promote self-evaluation of lifestyle behaviours. Suggested refinements of the manual’s design highlighted the importance of simplifying information and providing reassurance for patients early after a TIA or minor stroke. Information about fatigue, physical activity, and supporting goal setting was viewed as a key component of this novel secondary prevention initiative.

## 1. Introduction

Globally, acute cerebrovascular accident or stroke is a leading cause of disability and mortality [1] and its incidence increases with age [2]. Stroke events are frequently preceded by transient ischaemic attack (TIA) which affects approximately 50 in every 100,000 individuals per annum [2,3]. A 90-day cardiovascular event risk of approximately 20% has been reported following a TIA [4,5]. A TIA may be associated with more significant physical and mental health effects than has previously been recognized [6]. Therefore, early secondary prevention is recommended, [7] including behavioural interventions to reduce cardiovascular risk factors such as arterial hypertension and physical inactivity [8,9]. Systematic reviews have found limited evidence for the efficacy of such interventions in improvements of resting and peak systolic blood pressure, total cholesterol, aerobic capacity, or cardiovascular event rate mortality [10]. Further research is needed to identify an optimal approach to secondary prevention and effective promotion of physical activity, as well as reduction of sedentary behaviour in this population [11].

Following a TIA, individuals may experience a range of symptoms, anxiety about risk of recurrence and reticence regarding making behavioural changes such as increasing physical activity [11,12]. Cardiac rehabilitation has been associated with reductions in biological markers linked to both cardiovascular and cerebrovascular mortality [13]. Given that cardiac and cerebrovascular disease share common underlying mechanisms and risk factors, adapted physical activity-based, cardiac rehabilitation interventions could potentially be beneficial in the acute and sub-acute period after a TIA [14,15], however, their effectiveness is uncertain. Existing evidence indicates that aerobic exercise can result in important reductions in systolic and diastolic blood pressure in patients following a TIA, with effects most pronounced when exercise is initiated within six months of an initial event, and when interventions incorporate educational components [16]. A recent non-randomised study, examining cardiac rehabilitation undertaken in the sub-acute phase of recovery after stroke, found that participants took significantly more daily steps and engaged in more bouts of moderate or vigorous-intensity physical activity [17]. However, supervised cardiac rehabilitation programmes represent a relatively time and resource intensive approach that may not be sustainable or cost effective in current healthcare system settings. Poor initial uptake and long-term attendance, particularly among some groups such as older adults, may also limit the efficacy of such interventions [18].

Interventions which include pedometers have been shown to be effective for promoting increased physical activity levels in various populations [19,20]. Pedometers appear to be feasible for use by individuals after stroke [21] and can be used to promote increased walking activity, one of the commonest forms of physical activity that older adults engage in [22]. Their effect is thought to be mediated via various behaviour change techniques, including goal setting, provision of feedback, and self-monitoring [23,24]. Maintaining a lifestyle change is complex and relies on interactions between individual influences such as self-efficacy and environmental factors such as the method by which support is provided [25].

This study reports the initial stage of a larger body of work concerned with developing and testing a novel secondary prevention intervention for cardiovascular risk reduction in the early phase following a TIA or “minor” stroke. This intervention includes a paper-based manual, *“The Healthy Brain Rehabilitation Manual”* based on the content of the “Heart Manual” [26,27] and modified using findings from two systematic reviews [28,29] which identified behaviour change techniques relevant to patients with TIA or minor stroke, and the use of pedometers to promote physical activity. The manual was developed by clinicians and researchers with an interest in TIA, stroke, and cardiovascular research and is freely available through the current authors; it includes information relevant to stroke prevention, aimed at informing a lay population, and designed to promote healthy behaviours using specific behaviour change techniques. When developing any complex health care intervention, it is critical that it is acceptable to end users to maximise implementation in practice [30]. Therefore, the aim of this study was to assess the views of an expert group of patients and health professionals on the usability and acceptability of an initial draft of this intervention designed for secondary prevention in a TIA or minor stroke population, with the purpose of refining its design prior to a trial to determine its effectiveness.

## 2. Materials and Methods

### 2.1. Study Design

To explore stakeholder perspectives on the intervention, qualitative data were gathered using focus group discussions involving patients, carers, health professionals, and researchers with an interest in cardio/cerebrovascular disease or primary care. A researcher with experience in qualitative research involving patient groups [NH] conducted the focus groups between November 2015 and March 2016. Recommendations of the consolidated criteria for reporting qualitative research (COREQ) were followed [31]. The study was approved by the Office for Research Ethics Committees, Northern Ireland (NI) (REC reference 15/NI/0001, 21 September 2015).

### 2.2. Study Participants

Focus group participants were purposively selected to capture a range of views across clinicians, researchers, post-TIA patients, and a carer. Participants in the first focus group were recruited through the Chest, Heart and Stroke (NI) charity (4 patients (1 female, 3 males, 1 post-TIA, 3 post minor stroke) with one female carer, age range 29–76 years old). All 5 individuals who were contacted by the charity agreed to attend the focus group, which was conducted at the charity’s premises. For the second focus group, all health professionals (*n* = 9) working within an acute stroke unit in Belfast were invited via an open letter from the researcher. This focus group was held during a lunchtime period in the hospital’s stroke ward. It was comprised of 9 health professionals, all female (3 stroke nurses, 2 physiotherapists, a stroke consultant, pharmacist, occupational therapist, and speech and language therapist; age range 31–58 years old). Participants in these focus groups had no previous relationship with the focus group moderator.

A third focus group comprised public health researchers based in the Centre for Public Health, Queen’s University Belfast. Four participants were selected purposively because of their experience in intervention and service development and varied disciplinary backgrounds. All participants had experience working with TIA patients, stroke services and preventive health care. All participants agreed to participate (1 female, 3 males, one general practitioner, a social scientist-social worker, a public health medicine consultant, and a health psychologist, age range 30–63 years old). These participants had a professional working relationship with the moderator (NH).

Each group was informed that the purpose of the discussion was to explore their views of an initial version of “The Healthy Brain Rehabilitation Manual” with the aim of refining it to improve its usability and acceptability for patients following a TIA or minor stroke.

For this study, TIA and “minor” stroke are defined clinically by the patent’s history. For further research examining the effectiveness of the intervention, TIA and minor stroke will be defined by clinical history, a neurological examination and neuroimaging, with evidence of infarction indicating a diagnosis of stroke and the absence of infarction indicating TIA (266 Siket, Matthew S. 2012;). TIA is defined as “a transient episode of neurological dysfunction caused by focal brain, spinal cord or retinal ischaemia, without acute infarction” (267 Easton, J. Donald 2009;). A minor stroke is defined by a score of 3 or less on the National Institutes of Health stroke scale at initial assessment (1375 Fischer 2010;) (1376 Park, TH 2013;).

### 2.3. Data Collection

With their informed consent, all participants were emailed an electronic version of the manual to allow them time to review the content over a two-week period preceding the focus group. Table 1 illustrates the core components of the manual. Participants were also given a paper copy of it approximately 30 min before starting the focus group. Semi-structured schedules following a topic guide with open-ended questions were used to guide group discussions (see Table 2). These questions were based on previous research [32] and were reviewed, piloted, and adapted prior to the study commencing. All focus groups were conducted face-to-face, lasted approximately one hour, were audio-recorded, and transcribed verbatim. Field notes were recorded and were summarised to support the analysis and interpretation of data. No repeat interviews were conducted.

### 2.4. Data Analysis

A reflexive thematic analysis was used based on an inductive approach [33]. Transcribed data underwent a five-stage process during analysis. This included, data familiarisation, coding, generation of initial themes, review and definition of themes, and an analytical narrative synthesis to contextualise findings in relation to existing evidence. Data were coded by hand and 2 authors (N.H. and M.C.) cross-coded transcripts and met to discuss and resolve any disagreement. All decisions were discussed and confirmed via consensus among all authors.

## 3. Results

Use of a reflexive thematic analysis approach allowed for the views of each group to be ascertained on their relative strengths and limitations, as well as on possible areas for improvement of the tool. The analysis identified the following three overarching, inter-related themes: (1) relevant information and content, (2) accessible format and helpful structure, and (3) strategies to optimise usage and implementation in practice. These themes are described below.

### 3.1. Theme 1: Relevant Information and Content

Information in the manual was widely viewed as being comprehensive and relevant to patients following a TIA or minor stroke. Participants frequently suggested that the advice provided was central to increasing knowledge and awareness around the possible effects on the individual, and on the impact these factors could have on physical activity and other lifestyle behaviours. The information on medication and diet were seen as being particularly useful and both patients and health professionals regarded the information on fatigue as being especially important.

Some patients felt the impact of fatigue experienced post TIA was not widely understood or fully explained by some health professionals. One patient reflected that,
“I was telling him about the tiredness and fatigue and he told me that fatigue and tiredness has nothing to do with the stroke. I just got up and left, I didn’t even want to speak with him”(male, patient, 49 years)

Health professionals in the study also emphasised the importance of discussing fatigue and differentiating it from “normal tiredness”. They acknowledged failure to recognise the impact that this fatigue could have on patients might be compounded by a lack of health professional’s knowledge about this symptom.

“I would highlight to people that stroke fatigue is different from normal fatigue—this is not something which gets better with rest and sleep….”(female, physiotherapist, 37 years)

“The important thing to tell patients as well is that it (fatigue) is not always something which goes away or gets better but that it is about managing it…it also makes them feel more normal knowing that it might not get better….”(female, stroke nurse, 43 years)

Health professionals suggested that more information would be useful relating to self-management of fatigue symptoms, for example, by including information on “energy conservation”. Others highlighted the positive effect of the “psychosocial support” provided by the manual, to help address anxiety and mood changes. Participants highlighted the potential for the information about the importance of making lifestyle modifications and practical support on how to make changes, to help improve motivation and self-confidence, and to reassure patients that another cardiovascular event is “hopefully less likely to happen”. In terms of making changes to lifestyle behaviours, it was perceived that it was important that motivation was maintained, to maintain changes and avoid reversion to previous behaviours.

“…for me, it is hard to keep motivated. In the first few months you are out walking, you are careful with your diet. As time goes on there is a danger that you can just drift back into bad habits, same old, same old….”(male, patient, 64 years)

Other patients indicated they felt unsure about how much exercise and physical activity to do themselves. The inclusion of a pedometer with the manual was, therefore, typically seen as worthwhile, both in terms of a means of self-monitoring of daily steps, and as a way of prompting patients to start to be more physically active.

“I think if it says, look, this is what you need to do—try and do 7500 steps/day, use a pedometer. Then it is back to the individual to either do it or not. You know, putting the onus back on yourself and knowing what you have to do. Obviously for me it was about losing a bit of weight but I thought that was good…having that accountability....”(male, patient, 68 years)

Two health professionals suggested that the section on using home monitoring devices to record blood pressure could lead to increased anxiety if blood pressure was consistently high, but this view was not supported by others who considered that self-monitoring of blood pressure was an important element of self-regulation and control.

### 3.2. Theme 2: Accessible Format and Helpful Structure

The format of the manual was primarily regarded by participants as being useful for providing patients with an accessible source of information and support in the early stages after a TIA or minor stroke. The paper-based booklet was perceived as clear and simple to follow. Health professionals and patients indicated that it could promote patient self-monitoring and support self-evaluation of their early progress. Goal setting and the ability to use the manual to record physical activity targets was also seen as important. Health professionals highlighted the significance of providing reassurance to patients and reinforcing key messages in the early phase after a TIA that “might easily be forgotten”, for example, following discharge from an acute hospital setting. It was suggested that most of the information should be provided near or at the beginning of the manual to be most useful to patients. A number of patients suggested that having access to the manual could fill the gaps in care between hospital discharge and follow-up appointments that might sometimes only take place a number of weeks or months after discharge.

Some health professionals and researchers indicated that the tool was “very collaborative” and could both facilitate and enhance conversations between patients and health professionals, as well as highlight areas for discussion. Another positive aspect of the manual’s structure was the inclusion of “simple and clear” hints and tips. Some pointed to the usefulness of including tips on “simple, easy exercises that could be done in the house and which don’t need any equipment”, as well as tips relating to other lifestyle behaviours.

“I think the section on smoking is very useful. I’m an ex-smoker and I would have used most of the hints to help me stop. The hints are very good, to help you change your habits.”(female, patient, 70 years)

“…the other technique which applies to most of these sections is getting you to write down a ‘if when plan’. For example, if I feel hungry, take low sugar chewing gum….”(male, patient, 56 years)

Some patients and researchers stated that the brief format of the manual could allow it to be used as a general information booklet in (primary) cardiovascular risk prevention, advocating that it could be aimed at wider populations, including the general public and particularly at younger adults at increased cardiac risk. Others indicated that the booklet format was useful for helping to make family members view the material and make them more aware of the causes, consequences, and warning signs of a recurrent TIA or stroke. Participants in all groups highlighted the significance of social factors, and frequently referred to the possible involvement of family members and friends for providing support. For example,
“…yes, well my daughter read it and was taking in the information, you know, better diet and all the rest of it. So, she took an interest in it and in helping me….”(male, patient, 68 years)

### 3.3. Theme 3: Strategies to Optimise Usage and Implementation in Practice

Several participants felt that to improve long-term usage and engagement, the diary section of the manual should be expanded to facilitate greater use of goal setting techniques including realistic, progressive physical activity targets. Some indicated that this could also allow for goals and progress to be discussed more easily with health professionals, as well as with family or friends, and could provide support and motivation for patients to maintain physical activity changes over time. One patient highlighted an issue with accurate recording of pedometers based on previous experience and cited this as a potentially significant demotivating factor, emphasising the importance of providing reliable, easy to use pedometers. One health professional highlighted the importance of promoting other forms of physical activity, in addition to walking.

A possible barrier to usage that was identified was that the text was seen as being too long in some sections and areas were highlighted where more plain language could be used. Health professionals suggested that the manual should also signpost users to additional resources including information on local patient support groups or exercise and rehabilitation programmes.

Participants saw the manual as being valuable when used in different ways, with one patient stating that it would be more likely to be used on one or two occasions but could underpin some key messages.

“I have to admit I wasn’t running back to the manual too often but once the message was there, I think it was just trying to help yourself with this…The main messages I got from the book were a good diet and exercise more….”(male, patient, 56 years)

Others suggested that it was realistic and feasible for the manual to be used over the long term as a means of monitoring progress and recovery. When discussing its long-term use, it was frequently highlighted that additional health professional support and interaction would be beneficial and that “actually talking to or communicating with someone might be more powerful than just using it by itself”. One suggestion was that a section could be added for health professionals to “add notes”, for example, adding simple diagrams to illustrate different home exercises. These issues led participants to discuss how it might also be delivered in a digital format, including a mobile app, which may increase its reach and accessibility. A small number of health professionals raised possible technical or usability issues with apps as a barrier to use. However, participating patients were confident in using digital technologies and had experience using other mobile health apps. Some health professionals did emphasize that both paper-based and app versions should be available.

## 4. Discussion

The aim of this study was to assess the views of patients and health professionals on the usability and acceptability of an initial version of the “The Healthy Brain Rehabilitation Manual”, an innovative approach to secondary prevention following a TIA or minor stroke. Our findings provide valuable evidence relating to the information and content provided in the manual, its format and structure, and how its use could be optimised in practice. Participants’ perspectives were typically positive, particularly regarding the relevance of information provided about the symptoms and effects of the condition, the accessible format, and the helpfulness of practical advice and support provided, including its potential to promote self-evaluation of lifestyle behaviours. The inclusion of a pedometer for goal-setting and self-monitoring of physical activity was perceived as being important and suggestions were made to improve its readability and relevance for maintaining behaviour change.

While there is currently limited evidence for the effectiveness of lifestyle modification interventions in TIA and minor stroke populations, current guidelines for individuals at high cardiovascular risk recommend participation in at least 150 min of regular, moderate intensity aerobic activity per week [34] and a cardioprotective diet. However, physical activity levels have been shown to be relatively low in this population. In a recent systematic review of physical activity interventions post TIA [35] of eight included trials, three reported change in time spent in moderate-to-vigorous physical activity, but only one of these showed any significant increase in activity [36]. Our findings suggest that the manual had potential value for increasing awareness of the possible ‘invisible’ effects of a TIA, such as increased fatigue, and their consequent impact on individuals’ perceived ability to make lifestyle changes, including increased physical activity. An improved understanding of their physical condition may help individuals to engage in behaviour change programmes.

Our participants indicated that the manual could help to provide reassurance and address gaps in support provided early after a TIA or minor stroke. Practical advice on how to make lifestyle changes was also seen as a potential means to improve patients’ motivation and self-confidence, suggesting that the potential efficacy of the manual, used with a pedometer, may be based on behavioural control and self-efficacy, or perceived ability to carry out a given behaviour [37]. Increased self-efficacy can also result in more effective responses to barriers or setbacks when making lifestyle changes, with individuals more able to apply behaviour change maintenance strategies such as action planning [38]. Goal setting and recording physical activity targets was viewed as being important for supporting self-evaluation of progress based on measurable data. Interventions based on this theoretical underpinning have been associated with greater intention to change behaviour [38].

Manual pedometers were acceptable to study participants for self-monitoring of daily steps and promoting increased physical activity. Patients suggested that using the goal setting diary, and then discussing this with a health professional, was a particularly positive feature. This supports evidence from studies which have reported that participants valued pedometer feedback on walking activity during a cardiac telerehabilitation programme [39] and that using individual step goals increased motivation to be physically active and supported relatedness to others [39,40].

In terms of optimising implementation of the intervention, a key concept emerging from the data was the ability to deliver it as a digital intervention. Existing literature suggests that mobile app-based interventions can be associated with increased user engagement and have more effective outcomes as compared with other forms of supported self-management [40]. App-based interventions can provide evidence-based health information and incorporate adaptive methods including prompts and reminders [41]. Whilst many patients do not use apps for various reasons, including usability issues, lack of time, or lack of perceived relevance [42] digital interventions for supporting self-management and lifestyle modification can provide wider accessibility, good cost-effectiveness, and effectiveness in increasing physical activity among patients with chronic conditions, including cardiovascular disease [43,44,45,46,47]. Evidence evaluating mobile apps for supporting lifestyle modification in various clinical populations indicates that behavioural techniques including goal setting, performance monitoring, and feedback are common features [44].

Although some health professional participants cautioned against the advice on use of home blood pressure monitoring, there is evidence that self-monitoring may improve blood pressure management in primary care patients [48]. Because blood pressure is a key modifiable risk factor for recurrent stroke, further studies are needed to explore potential barriers and facilitators to using such interventions in this population.

### 4.1. Strengths and Limitations

This study has a number of strengths. The qualitative data gathered provide a valuable evaluation of views on the acceptability of the tool, based on the perspectives of patients and health professionals with a range of differing characteristics and experiences. Similar issues were raised in all focus groups. The use of separate focus groups for patients and health professionals appeared to allow all participants to speak freely and openly, which may not happen in mixed groups. The study yielded information which has been used to modify our intervention (see Table 3). However, participating patients were recruited via a patient support group, and therefore may be more aware of the implications and risks of TIA and stroke, so that they may not be fully representative of typical patients. Focus groups were facilitated by a researcher who had involvement in the design of the manual. However, this potential source of bias and influence on participant responses was minimised by the use of a topic guide, audio-recording, the reflexive approach undertaken, cross coding of data, and use of consensus decisions during analysis and interpretation of the study findings. Further development of this complex health services intervention will be informed by feasibility testing of the use of the revised manual among patients with recent TIA or minor stroke (four weeks post-event) and explorative research with triangulation of methods of data collection.

### 4.2. Conclusions

Findings indicate that “The Healthy Brain Rehabilitation Manual” is an acceptable approach to secondary prevention, with promotion of physical activity, in the early phases following a TIA or minor stroke. Some key improvements have been suggested, to optimise its use and implementation in routine practice. These include a reduction of text, greater use of plain language, more information on physical activity, with an expanded physical activity and goal-setting diary and sign posting to programmes which support long-term lifestyle modifications. These modifications have been made, and a revised version has been pilot tested with 40 patients [49,50]. Our study has added to evidence of how interventions for secondary prevention should consider individuals’ needs, provide evidence-based advice, and relevant support and resources. Further work is being conducted to develop an app-based version of this novel manual for the TIA/minor stroke population.

## Figures and Tables

**Table 1 ijerph-17-08788-t001:** Core components of “The Healthy Brain Rehabilitation Manual”.

Introduction
- Goal setting - Action planning - Examples of action planning - Goals and action plan diary - Information on recognising signs of a TIA * or stroke
Section 1: Stopping smoking
- Benefits of stopping smoking - Preparing to quit - Hints and tips to help stop smoking
Section 2: Be more active, more often
- Benefits of physical activity - What is exercise and physical activity? - Recommended amount of exercise and physical activity? - Warming up before exercise and physical activity - Exercises to try at home - Exercise and physical activity diary
Section 3: Having a healthy diet
- Recommended diet - Setting targets
Section 4: Managing stress
- What is stress? - Hints and tips for managing stress
Section 5: Managing fatigue
- Hints for managing fatigue - Other treatments for fatigue
Section 6: Medication
Section 7: Community resources
Section 8: General information on TIA and minor stroke

* TIA: Transient Ischaemic Attack.

**Table 2 ijerph-17-08788-t002:** Topic guides for interviews and focus group discussions.

Questions
- What do you think of the manual? How often did you use or refer to it? Was there anything you found hard to understand or would want more information on? - What do you think about the general level of information included—too much/too little? - What do you think of the length of the booklet? (short/long)? - What do you think of the layout of the booklet? (colour, pictures, font size)? - What did you think of the patient story? - Did family members/friends read the manual? What did they think? - Anything which you particularly enjoyed or disliked? Anything you would change in it? - If an app version was available, would people use it? - How do you feel about exercise and physical activity after having had a TIA/stroke? - What do you think of the exercise and physical activity programme? Could you do this in your own home? Is it challenging? - What are your views on using the pedometer/step-count goals? - What would you think of using an electronic aid, e.g., Fitbit, app on phone to help you get more active? - Are there other ways of follow-up which you would like to see included, e.g., group meetings, on-line chat rooms?

**Table 3 ijerph-17-08788-t003:** Key changes to the “The Healthy Brain Rehabilitation Manual” based on qualitative data.

Change Made
- Addition of a blank section to allow patients to write down questions to ask health professionals at their next appointment. - Additional information included on physical activity and exercise advice and ‘signposting’ to local rehabilitation programmes and other community resources. - Additional space added for the physical activity diary. - Volume of text reduced in sections. - Text altered in sections to include more user-friendly language.

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
