# Peer review of "Usability and Acceptability of a Novel Secondary Prevention Initiative Targeting Physical Activity for Individuals after a Transient Ischaemic Attack or “Minor” Stroke: A Qualitative Study"

_ijerph, 2020, doi:10.3390/ijerph17238788_

Round 1

Reviewer 1 Report

I question the purpose of the study - the Authors did not fully present for whom was prepared the manual / guide ? It is also unknown where and under what circumstances it was, is and will be available, and whether it is free or paid? Does the handbook cover basic or extended knowledge? Who is the author of this manual / quide (what specialist?
It is also unknown how important are this research and its results for the wider community. What is the range for the results of this study?

Please explain a few facts related to the choice of focus groups:
1) size and such a large diversity of groups, including diversity of age groups (especially in the first focus group),
2) representativeness of the groups - has it been specified in some way?
3) legitimacy of selecting 3 focus groups.

Author Response

Reviewer 1

I question the purpose of the study - the Authors did not fully present for whom was prepared the manual / guide ?

The purpose of the study was to explore the views of stakeholders, including patients, carers, health professionals and researchers, on an initial draft of the manual which is the key component of a novel intervention to promote secondary prevention after a TIA or minor stroke. Its development has followed MRC guidance for development and evaluation of a complex health services intervention and this study reports the initial stage of its development. It was developed for use by people following a TIA and ‘minor’ stroke, as stated within the title, introduction, methods and discussion of the paper. We have revised the final paragraph of the introduction to hopefully clarify these points and specifically stated the purpose ‘of refining its design prior to a trial to determine its effectiveness.’ in the final sentence of introduction.

It is also unknown where and under what circumstances it was, is and will be available, and whether it is free or paid?

We have added to the paper, in the final paragraph of the introduction:

“The manual was developed by clinicians and researchers with an interest in TIA, stroke and cardiovascular research and is freely available through the authors.”

Does the handbook cover basic or extended knowledge? Who is the author of this manual / quide (what specialist?

The content of the manual is included within Table 1. The manual covers extended knowledge related to secondary cardiovascular prevention.

It is also unknown how important are this research and its results for the wider community. What is the range for the results of this study?

The introduction details the importance of this research in that early intervention following a TIA or minor stroke is key to reducing the risk of further recurrence of cardiovascular or cerebrovascular events. As yet there is a lack of evidence for an effective approach to optimising patients’ engagement in secondary prevention, including both behaviour change and medication. The results of this study have informed the refinement of the manual for the next stage in its development and evaluation. The results also have identified issues perceived by key stakeholders, both professional and lay, as important in engaging with this population group and may be extrapolated to interventions in other clinical conditions. The range for the results is therefore wide, having significance to others working in this field of research as well as the target group of patients within the population.  

Please explain a few facts related to the choice of focus groups:
1) size and such a large diversity of groups, including diversity of age groups (especially in the first focus group).
2) representativeness of the groups - has it been specified in some way?
3) legitimacy of selecting 3 focus groups.

The ideal size of focus groups is 5 to 8 participants: our groups ranged in size from 4 to 9. It is difficult to recruit exact numbers as the time/ location chosen may become unsuitable for some potential participants. We wished to include a large diversity of characteristics, including age, within our groups in order to explore as wide a range of views as possible, given time constraints for our work. We chose the three focus groups to reflect the views of stakeholders whom we considered relevant – that is, TIA and ‘minor’ stroke patients and their carers, health professionals and researchers.

Reviewer 2 Report

This study assessed the views of patients and health professionals on the usability and acceptability of an initial version of the ‘The Healthy Brain Rehabilitation Manual’, an innovative approach to secondary prevention following a TIA or ‘minor’ stroke.

  1. What is the definition of TIA and minor stroke?
  2. What type of stroke etiology were the stroke patients included in the study?
  3. How was the initial stroke severity (ie. NIHSS score) in stroke patients?
  4. How long did it take from the stroke onset to the interview with the stroke patient?
  5. On line 132 and 189, you need to clear the typo ".".
  6. Medication adherence for secondary prevention is closely related to stroke recurrence and prognosis. If there is a section in the Manual to check the medication adherence, it may be more effective in preventing stroke recurrence. What do the authors think about this?
  7. What tips and methods do the authors think to reasonably set physical activity goals in patients with TIA/minor stroke?
  8. For example, what are some of the expanded goal-setting component?
  9. When the authors develop the app-based version of the patient guide or aid, are there any newly designed functions and features to maintain lifestyle change well, unlike other mobile apps?

Author Response

Reviewer 2

This study assessed the views of patients and health professionals on the usability and acceptability of an initial version of the ‘The Healthy Brain Rehabilitation Manual’, an innovative approach to secondary prevention following a TIA or ‘minor’ stroke.

 What is the definition of TIA and minor stroke?

The patients targeted by this intervention will have experienced a TIA or ‘minor’ stroke as defined clinically by the patent’s history, a neurological examination and neuroimaging (typically a CT and/or MRI head scan), with evidence of infarction indicating a diagnosis of stroke and the absence of infarction indicating TIA{{266 Siket, Matthew S 2012;}}. TIA is defined as “a transient episode of neurological dysfunction caused by focal brain, spinal cord or retinal ischaemia, without acute infarction” {{267 Easton, J. Donald 2009;}}. Whilst stroke has traditionally been defined as “an acute onset of focal neurological symptoms which last more than twenty-four hours” {{875 Warlow, C 2008;}}, a more up-to-date definition is “the acute onset of focal neurological symptoms from the brain, retina or spinal cord of any duration, in which imaging or autopsy show focal infarction or haemorrhage relevant to the symptoms” {{876 Sacco,Ralph L. 2013;}}. ‘Minor’ stroke is defined by a score of 3 or less on the National Institutes of Health stroke scale at initial assessment as per previous authors {{1375 Fischer 2010;}} {{1376 Park, TH 2013;}}.

  1. What type of stroke etiology were the stroke patients included in the study?

The patients in the current study self-reported their clinical conditions as TIA or minor stroke; for the purpose of this initial work we did not seek confirmation of stroke etiology. For the next stage of our work this will be defined as indicated above.

  1. How was the initial stroke severity (ie. NIHSS score) in stroke patients?

Please note response above; for the next stage in our work we define stroke severity using the guidance that ‘Minor’ stroke is defined by a score of 3 or less on the National Institutes of Health stroke scale at initial assessment {{1375 Fischer 2010;}} {{1376 Park, TH 2013;}}.

  1. How long did it take from the stroke onset to the interview with the stroke patient?

This varied and no specific time limit was used in the current study.

  1. On line 132 and 189, you need to clear the typo ".".

Corrected thank you, and we have also corrected other typos in the manuscript.

  1. Medication adherence for secondary prevention is closely related to stroke recurrence and prognosis. If there is a section in the Manual to check the medication adherence, it may be more effective in preventing stroke recurrence. What do the authors think about this?

Yes, the manual is based on the principles of cardiac rehabilitation and therefore tries to address as many behaviour change approaches as possible which will aid in secondary cardiovascular prevention, including medication adherence. Section 6 in the manual addresses medication adherence. (see Table 1 for manual components)

  1. What tips and methods do the authors think to reasonably set physical activity goals in patients with TIA/minor stroke?

The important thing with goals is to ensure they are patient centred and the best approach to this is through patient education, allowing the patient to understand the risks following a TIA and ‘minor’ stroke and letting them set appropriate physical activity guidelines for themselves. This is at the heart of the manual’s approach.

  1. For example, what are some of the expanded goal-setting components?

See point above. The manual is about educating the patient about their cerebrovascular event, allowing them to understand the risks and how their behaviours may have contributed to their overall risk of developing this event and indeed, their future risk of developing further cardiovascular events. From this position of education, the patient is encouraged to set their own targets and goals, discussing this with the stroke nurse (facilitator) and friends/family. The friends/family are also encouraged to participate in these targets and goals, helping the patient to achieve them. The refined version of the manual allows more space for patients to record goals and their progress towards meeting these.

  1. When the authors develop the app-based version of the patient guide or aid, are there any newly designed functions and features to maintain lifestyle change well, unlike other mobile apps?

The most important aspect of this intervention is that it is designed by patients, for patients and this will help with compliance. Also, targets are self-set by patients, included within the App and can then be reviewed by the patient and family as they enter new data for the target, e.g. pedometer step counts, blood pressure readings. We are currently preparing a paper focused on our App design.

Reviewer 3 Report

Logic is easy to follow and qual data presented in an easy flow fashion

Author Response

Reviewer 3:

Logic is easy to follow and qual data presented in an easy flow fashion

Thank for the review.

Reviewer 4 Report

COMMENTS AND SUGGESTIONS FOR AUTHORS

Title: The title should be modified, not including words like informed theory

Abstract: The abstract must be improved, beacuse it does not summarize the main ideas of the article.

In the introduction as in material and methods section, the secondary prevention intervention should be explained more

Study participants are unclear: cardiovascular disease?

The authors must specify the time taken by participants to read the manual

The level of studies of the participants was considered. This aspect should have been taken into account

In patients, the time elapsed since the event was taken into account

The age range of all participants must be specified

It would have been interesting to use more than one method for data collection, triangulating them

In the data analysis, as was done in the absence of consensus

Author Response

Reviewer 4:

Title: The title should be modified, not including words like informed theory

This has been modified.

Abstract: The abstract must be improved, because it does not summarize the main ideas of the article.

Thank you for this comment - we have revised the abstract to include more information.

In the introduction as in material and methods section, the secondary prevention intervention should be explained more

The content of the manual has been described in Table 1 and the intervention has also been previously published:

Previous articles published on the intervention, including both a feasibility and pilot trial:

  • Stroke Prevention Rehabilitation Intervention Trial of Exercise (SPRITE) - a randomised feasibility study.Heron N, Kee F, Mant J, Reilly PM, Cupples M, Tully M, Donnelly M. BMC Cardiovasc Disord. 2017 Dec 12;17(1):290. doi: 10.1186/s12872-017-0717-9. PMID: 29233087 Free PMC article. Clinical Trial.
  • Rehabilitation of patients after transient ischaemic attack or minor stroke: pilot feasibility randomised trial of a home-based prevention programme.Heron N, Kee F, Mant J, Cupples ME, Donnelly M.Br J Gen Pract. 2019 Sep 26;69(687):e706-e714. doi: 10.3399/bjgp19X705509. Print 2019 Oct. PMID: 31501165Free PMC article. Clinical Trial.

Study participants are unclear: cardiovascular disease?

The methods section clearly lays out the characteristics of the participants of the focus groups. The study participants who were patients previously had a TIA and/or stroke. The health professionals were those who worked with patients who had a stroke or in TIA clinics. The researchers were involved in health services research and all had experience of research work with patients with cardiovascular and cerebrovascular disease and its prevention. We have used the broad term cardiovascular to include cerebrovascular as both conditions have similar modifiable risk factors.

The authors must specify the time taken by participants to read the manual

The method section clearly states that the participants had 30 minutes to review the manual prior to the focus groups.

The level of studies of the participants was considered. This aspect should have been taken into account

We recognise that the educational level of the participants is important and have considered this in publications of feasibility and pilot studies of this intervention. However, having regard to preserving anonymity of patients’ data in the current study we did not report their educational level: their comments emphasised the importance of revising some of the manual’s text to improve its readability.

In patients, the time elapsed since the event was taken into account

For this qualitative work, no. We just wanted feedback on patients’ views of the manual at this stage. For our later feasibility and pilot work, all participants were recruited within 4 weeks of their event.

The age range of all participants must be specified

The age range of all participants is included in the ‘Method’ section of the manuscript.

It would have been interesting to use more than one method for data collection, triangulating them in the data analysis, as was done in the absence of consensus

This paper focuses purely on the qualitative review of the initial intervention. The triangulation of data, with quantitative data, occurred in the feasibility and pilot studies. We have referred to this in the discussion of our limitations.

Reviewer 5 Report

This paper reports a qualitative component of a larger study, that focused on how stakeholders perceived an educational and physical activity promotion intervention. While the title of this paper suggested that this paper was reporting a "theory informed intervention", there was no reference to a theory that framed this research. Was there an actual theory guiding this research, such as a patient educational, self management, self efficacy theory used?  This would have strengthened the introduction in supporting the rationale of this study, and a brief discussion of the theory incorporated into methods would have set the stage for how the theory was integrated into qualitative data collection and analysis.  If there was not a theory guiding the research, the title needs to be adjusted. 

A reflexive thematic analysis was cited, and was described, and use of the COREQ criteria was stated. The manuscript would be stronger if the COREQ criteria checklist were provided as an appendix or table of the manuscript to make the methods more transparent. Three overarching themes were presented in results, which could have been more fully fleshed out. For example, Theme 1: Information and content provided; the theme itself is not stating the impact of this content provision. The first sentence in the description of the theme stated it, and it would be far more meaningful to add a few more words to the theme. Same for Theme 2: Format and structure; what about the format and structure?  It is in the first sentence again describing the theme. Theme 3 was stated more effectively.  Results and representative quotes were well presented otherwise. 

Discussion and conclusion drew from the literature and were comprehensive. Overall, the paper would benefit from revision to address the above concerns.

Author Response

Reviewer 5

This paper reports a qualitative component of a larger study, that focused on how stakeholders perceived an educational and physical activity promotion intervention. While the title of this paper suggested that this paper was reporting a “theory informed intervention”, there was no reference to a theory that framed this research. Was there an actual theory guiding this research, such as a patient educational, self management, self efficacy theory used?  This would have strengthened the introduction in supporting the rationale of this study, and a brief discussion of the theory incorporated into methods would have set the stage for how the theory was integrated into qualitative data collection and analysis.  If there was not a theory guiding the research, the title needs to be adjusted. 

Thanks for this comment. Whilst we have used specific behaviour change techniques in designing the manual, and discuss the relevance of these in our discussion, we have not modelled it on a specific theory and thus the title has been amended.

A reflexive thematic analysis was cited, and was described, and use of the COREQ criteria was stated. The manuscript would be stronger if the COREQ criteria checklist were provided as an appendix or table of the manuscript to make the methods more transparent.

The COREQ checklist has been added as an Appendix.

Three overarching themes were presented in results, which could have been more fully fleshed out. For example, Theme 1: Information and content provided; the theme itself is not stating the impact of this content provision. The first sentence in the description of the theme stated it, and it would be far more meaningful to add a few more words to the theme. Same for Theme 2: Format and structure; what about the format and structure?  It is in the first sentence again describing the theme. Theme 3 was stated more effectively.  Results and representative quotes were well presented otherwise. 

Thank you for these suggestions which are very helpful and we have revised our themes accordingly. We now report Theme 1 as ‘Relevant information and content’ and Theme 2 as ‘Accessible format and helpful structure’.

Discussion and conclusion drew from the literature and were comprehensive. Overall, the paper would benefit from revision to address the above concerns.

Thank you.

Round 2

Reviewer 1 Report

Thank you for answering the questions. I wish the Authors good luck in further research and in various educational activities in the secondary prevention of patients with cardiovascular diseases.

This manuscript is a resubmission of an earlier submission. The following is a list of the peer review reports and author responses from that submission.